# Exploring the diagnostic potential of adding T2 dependence in diffusion-weighted MR imaging of the prostate

Ingrid Framås Syversen[1]*, Mattijs Elschot[2,3], Elise Sandsmark[3], Helena Bertilsson[4,5], Tone Frost Bathen[2], Pål Erik Goa[6]

1 Kavli Institute for Systems Neuroscience, NTNU, Norwegian University of Science and Technology, Trondheim, Norway, 2 Department of Circulation and Medical Imaging, NTNU, Norwegian University of Science and Technology, Trondheim, Norway, 3 Department of Radiology and Nuclear Medicine, St. Olavs Hospital, Trondheim University Hospital, Trondheim, Norway, 4 Department of Urology, St. Olavs Hospital, Trondheim University Hospital, Trondheim, Norway, 5 Department of Clinical and Molecular Medicine, NTNU, Norwegian University of Science and Technology, Trondheim, Norway, 6 Department of Physics, NTNU, Norwegian University of Science and Technology, Trondheim, Norway

* ingrid.f.syversen@ntnu.no

## Abstract

### Background

Magnetic resonance imaging (MRI) is essential in the detection and staging of prostate cancer. However, improved tools to distinguish between low-risk and high-risk cancer are needed in order to select the appropriate treatment.

### Purpose

To investigate the diagnostic potential of signal fractions estimated from a two-component model using combined T2- and diffusion-weighted imaging (T2-DWI).

### Material and methods

62 patients with prostate cancer and 14 patients with benign prostatic hyperplasia (BPH) underwent combined T2-DWI (TE = 55 and 73 ms, b-values = 50 and 700 s/mm²) following clinical suspicion of cancer, providing a set of 4 measurements per voxel. Cancer was confirmed in post-MRI biopsy, and regions of interest (ROIs) were delineated based on radiology reporting. Signal fractions of the slow component (SF_slow) of the proposed two-component model were calculated from a model fit with 2 free parameters, and compared to conventional bi- and mono-exponential apparent diffusion coefficient (ADC) models.

### Results

All three models showed a significant difference ($p<0.0001$) between peripheral zone (PZ) tumor and normal tissue ROIs, but not between non-PZ tumor and BPH ROIs. The area under the receiver operating characteristics curve distinguishing tumor from prostate voxels was 0.956, 0.949 and 0.949 for the two-component, bi-exponential and mono-exponential

**Data Availability Statement:** The minimal dataset underlying this study is available in the paper and its Supporting Information. Imaging data cannot be shared publicly because the data contain potentially

identifying or sensitive patient information. Access to experimental data are available through the research administration manager at NTNU (Norwegian University of Science and Technology), kontakt@isb.ntnu.no.

**Funding:** - IFS: Norwegian University of Science and Technology, https://www.ntnu.no/ (project number 81850040). - ME: The liaison Committee between the Central Norway Regional Health Authority and the Norwegian University of Science and Technology, https://helse-midt.no/samarbeidsorganet (grant number 90265300). - TFB: The Research Council of Norway, https://www.forskningsradet.no/ (grant number 295013). The funders had no role in study design, data collection and analysis, decision to publish, or preparation of the manuscript.

**Competing interests:** The authors have declared that no competing interests exist.

models, respectively. The corresponding Spearman correlation coefficients between tumor values and Gleason Grade Group were fair (0.370, 0.499 and -0.490), but not significant.

## Conclusion

Signal fraction estimates from a two-component model based on combined T2-DWI can differentiate between tumor and normal prostate tissue and show potential for prostate cancer diagnosis. The model performed similarly to conventional diffusion models.

## Introduction

Magnetic resonance imaging (MRI) has been essential in the detection and staging of prostate cancer for several years [1, 2]. Different sequences are performed, where T2-weighted and diffusion-weighted imaging (DWI) constitute the basis of such a diagnostic protocol [3]. Prostate cancer is usually detected as homogeneous moderately hypointense focal areas on T2-weighted images, with a relatively low apparent diffusion coefficient (ADC). The Prostate Imaging-Reporting and Data System (PI-RADS) guidelines are used to detect clinically significant cancer, based on a combination of DWI, T2-weighted imaging and also dynamic contrast-enhanced (DCE) MRI [3]. For peripheral zone (PZ) cancers, DWI is the dominant factor for determining the PI-RADS score, while for non-PZ cancers, T2-weighted images are predominantly used. However, despite these standardized guidelines, the accuracy of detecting and staging clinically significant cancer is still variable, and overtreatment is a major problem [4–6]. Improved diagnostic tools are needed in order to better stratify patients to active surveillance or radical treatment. In addition, it can be challenging to separate between non-PZ cancers and benign prostatic hyperplasia (BPH) with low ADC, as these have similar imaging characteristics [7].

Low ADC in the prostate is commonly interpreted as restricted diffusion due to densely packed cells in tumor tissue [8]. However, this simplification does not consider the different mechanisms of the underlying tissue microstructure. One suggested extension to this is the bi-exponential model, which consists of a slow diffusion component representing the restricted diffusion within cells, and a fast diffusion component representing extracellular water [9]. This model has shown promising results in previous studies [10, 11].

Another common assumption is that T2 values and ADCs are independent of each other. However, studies have shown an interdependence of these parameters which appears to differ between tumor, normal prostate tissue and BPH [12–14], which could potentially be exploited for diagnosis. By performing DWI at different echo times (TE), it is possible to isolate the signal from subvoxel populations of water molecules with specific paired T2 values and ADCs associated with different components of the prostate. A three-component model using this principle has been suggested [15]. However, for such a model to be clinically feasible, it needs to have a relatively short acquisition time and a low computational cost. A simpler two-component model fulfills these requirements and would be consistent with a simple representation of the prostate: Water in the glandular lumen with long T2 and a high ADC, and water inside the cells with a shorter T2 and lower ADC [12].

In this work, we estimate signal fractions in a slow and a fast diffusion component using combined T2- and diffusion-weighted imaging (T2-DWI), and compare these between tumor, normal tissue and BPH in order to investigate the diagnostic potential of the model.

## Materials and methods

### Patients

76 patients underwent an extended MRI exam as part of the integrated cancer care pathway following prostate cancer suspicion. Of these, 62 patients had post-MRI biopsy-confirmed cancer and were assigned randomly to a training and a test set (see Table 1 for details). The inclusion criteria were tumor in any prostate zone that both had a PI-RADS and a location-matched Gleason score. The 14 included patients without detected cancer had BPH lesions, characterized by visually low ADC and a negative biopsy. An overview of the patient and case selection process can be found in the S1 Fig. The study was approved by the Regional Committee for Medical and Health Research Ethics Central Norway (identifier REK 2017/520), and all participants provided written informed consent before enrollment.

### MRI protocol

Imaging was performed on a 3T MRI scanner (Magnetom Skyra, Siemens Medical Systems, Erlangen, Germany) using body surface coils. The combined T2-DWI was added at the end of a clinical protocol.

**Table 1. Summary of patient characteristics for the 62 included patients with biopsy-confirmed cancer.**

| Parameter | Training set | Test set |
|---|---|---|
| Number of patients | 31 | 31 |
| PSA level (ng/mL) | 11.0 ± 8.6 | 11.4 ± 15.7 |
| PI-RADS score | | |
| 2 | 2 | 2 |
| 3 | 7 | 5 |
| 4 | 8 | 6 |
| 5 | 14 | 18 |
| Gleason Grade Group | | |
| 1 | 6 | 4 |
| 2 | 7 | 15 |
| 3 | 10 | 5 |
| 4 | 4 | 3 |
| 5 | 4 | 4 |
| Cancer location | | |
| PZ | 25 | 24 |
| TZ | 3 | 4 |
| CZ | 1 | 0 |
| AFMS | 2 | 3 |
| Treatment | | |
| RARP | 15 | 14 |
| Radiation therapy | 8 | 7 |
| Hormone therapy | 2 | 1 |
| Active surveillance | 6 | 9 |

PSA = prostate-specific antigen, PI-RADS = Prostate Imaging-Reporting and Data System, PZ = peripheral zone, TZ = transition zone, CZ = central zone, AFMS = anterior fibromuscular stroma, RARP = robotic assisted radical prostatectomy.

Note—Data are numbers of patients, except for PSA level which is given as mean ± standard deviation. There were 2 and 7 missing PSA values in the training and test set, respectively. The Gleason Grade Group is based on biopsy scores after MRI, except for the patients who underwent RARP, where the histopathological Gleason Grade Group was used. Note that the tumors might extend over multiple prostatic zones, and that the cancer location denoted is the primary tumor location.

The transversal combined T2-DWI acquisition consisted of two fat-suppressed, single-shot, monopolar spin-echo echo-planar imaging (EPI) sequences with TE = 55 and 73 ms, respectively. Each of these sequences had repetition time (TR) = 4200 ms, b-values = 50 and 700 s/mm$^2$ (three directions; number of excitations (NEX) = 2 and 4, respectively), resolution = 2.0×2.0×3.0 mm$^3$, field of view = 256×256 mm$^2$, imaging and reconstruction matrix = 128×128, 26 slices, generalized autocalibrating partial parallel acquisition (GRAPPA) factor 2 and acquisition time 1:38 minutes. The only differences between the sequences at the two TEs were the diffusion times and gradient amplitudes: at TE = 55 ms, δ = 11.6 ms and Δ = 23.9 ms, while at TE = 73 ms, δ = 20.6 ms and Δ = 32.9 ms. This protocol provides a 2×2 matrix of trace-weighted diffusion measurements for each voxel, where each measurement is associated with a different combination of TE and b-values.

## Preprocessing

All analyses were performed using MATLAB (version R2019b, MathWorks, Natick, MA, USA) unless stated otherwise. Code used for model fitting is available on GitHub at https://github.com/ntnu-mr-cancer/T2-DWI.

The trace-weighted images at each TE and b-value were co-registered to the image with the lowest TE and b-value with Elastix, using a multiresolution rigid registration scheme [16, 17]. The scanner's autogenerated ADC map for TE = 73 ms was also co-registered to the same image because the regions of interest (ROIs) were to be delineated on this map. To only correct for potential motion of the prostate and not of other internal structures, a box-shaped ROI covering the prostate was defined for each patient and used as a mask for the co-registration.

**ROI delineation.** For each cancer patient, one tumor ROI was manually delineated using ITK-SNAP (www.itksnap.org) [18] on the scanner's autogenerated ADC map for TE = 73 ms, with corresponding T2-weighted images used for support. The delineation was performed based on the clinical radiologist's PI-RADS annotation and the tumor ROI was characterized by focal low ADC under (or around) 1000 μm$^2$/ms. All tumor ROI locations were confirmed to be cancer by matching with biopsy reports in the patient journal, and cross-checked with histology slides if available (n = 29). For the PZ tumor patients, one normal tissue ROI was also delineated, characterized by visually high ADC in the PZ (around 2000 μm$^2$/ms). For both the tumor and normal tissue ROIs, respectively, only one tumor lesion or normal area were considered per ROI. For each BPH patient, one ROI was manually delineated of one or more proliferative BPH nodule(s) with visually low ADC (under/around 1000 μm$^2$/ms) in the non-PZ, also visible as a nodule on T2-weighted images. All ROIs were delineated by a basic scientist (IFS, 1 year of experience in prostate MRI) and validated by a radiology resident (ES, 1 year of experience in prostate MRI, supervised by an experienced radiologist).

## Two-component model

We modeled the MR signal as water in two separate components: a slow diffusion component with low ADC and short T2, and a fast component with high ADC and long T2. Thus, the 2×2 matrix of signal intensities SI from the combined T2-DWI were fitted to the following equation:

$$\frac{SI}{SI_0} = SF_{slow}\ exp\left(-\frac{TE}{T2_{slow}}\right) exp(-b * ADC_{slow}) + SF_{fast}\ exp\left(-\frac{TE}{T2_{fast}}\right) exp(-b * ADC_{fast}), \quad (1)$$

where the subscripts "slow" and "fast" denote the values of slow and fast components, respectively. $SI_0$ is the signal intensity at TE = 0 and b = 0, SF is the signal fraction of the components, and

$SF_{slow}+SF_{fast} = 1$. In order to reduce the number of free parameters in the model, $ADC_{slow}$ = 0.3 μm$^2$/ms and $ADC_{fast}$ = 2.6 μm$^2$/ms globally optimized for a biophysically similar bi-exponential model were used [19]. This results in four free parameters ($SI_0$, $SF_{slow}$, $T2_{slow}$, $T2_{fast}$) to the four measurements.

$T2_{slow}$ and $T2_{fast}$ were then globally optimized for the entire population of voxels across all patients in the training set by minimizing a global cost function while fitting the 2×2 signal to Eq 1 using a range of T2 values determined from a previous preliminary study [20]. The cost function was defined as the sum of the root-mean-square error (RMSE) of the fit of all included voxels. By keeping $T2_{slow}$ and $T2_{fast}$ fixed for each iteration, only two parameters were fitted for each voxel in this process. The optimal T2 values were then used for further analysis, where the two remaining free parameters $SI_0$ and $SF_{slow}$ were determined on a voxel-by-voxel basis by fitting the 2×2 signal to the two-component model for all included patients.

All voxels inside the box-shaped ROIs were analyzed. The average size of the box ROIs was approximately 167,000 voxels. However, to reduce noise effects, voxels were excluded that had a value equal to or below three times the noise floor, defined as the average signal intensity of background voxels. Voxels with an apparent negative ADC or T2 value were also excluded. On average, approximately 10% of the voxels in the tumor, normal and BPH ROIs were excluded.

## Bi-exponential model

For comparison, we also investigated a purely ADC-dependent bi-exponential model:

$$\frac{SI}{SI_0} = SF_{slow}\ exp(-b*ADC_{slow}) + SF_{fast}\ exp(-b*ADC_{fast}), \tag{2}$$

where $SF_{slow}+SF_{fast} = 1$, $ADC_{slow}$ = 0.3 μm$^2$/ms and $ADC_{fast}$ = 2.6 μm$^2$/ms as in the two-component model [19]. $SI_0$ and $SF_{slow}$ were fitted to the two b-value measurements at TE = 73 ms.

## Mono-exponential ADC

Using

$$\frac{SI}{SI_0} = exp(-b*ADC), \tag{3}$$

$SI_0$ and ADC were fitted to the two b-value measurements at TE = 73 ms.

An extended analysis with even more model comparisons can be found in the S1 Appendix.

## Statistical analysis

The first part of the statistical analysis was divided into PZ and non-PZ tumors. Note that all PZ analyses were performed on the test set only, while the non-PZ analyses were carried out on all available patients due to the low sample size. For the PZ analyses, the Wilcoxon signed-rank test was used to test for statistical significance between mean $SF_{slow}$ (for both the two-component and bi-exponential models) and ADC of the tumor and normal tissue ROIs (n = 24). For the non-PZ analyses, BPH ROIs from the BPH patients (n = 14) were used for comparison with the non-PZ tumor ROIs (n = 13), and the Mann-Whitney U test was used to test for statistical significance between these. All tests were two-sided. After a Bonferroni correction for 9 multiple comparisons (including the correlation described in the following paragraph), $p < 0.006$ was considered statistically significant.

The remainder of the statistical analyses were carried out on the whole test set with both PZ and non-PZ tumors together. Voxel-wise receiver operating characteristics (ROC) analysis

was performed comparing the $SF_{slow}$ and ADCs in the tumor ROIs with the rest of the area inside the box-shaped ROIs. Note that although the ROC analysis was performed only on the test set, the optimal threshold value was calculated from the training set and applied on the test set in the calculation of sensitivity and specificity. Furthermore, the Spearman correlation coefficient ($\rho$) was calculated between the mean $SF_{slow}$ and ADC of the tumor ROIs and the Gleason Grade Group.

## Results

The optimal T2 values for the two-component model were determined to be $T2_{slow} = 45$ ms and $T2_{fast} = 180$ ms (Fig 1).

Box plots of estimated $SF_{slow}$ (for both the two-component and bi-exponential models) and ADC for different ROIs are shown in Fig 2. In the PZ analyses, all metrics show a significance between tumor and normal ROIs. In the non-PZ analyses, no metrics show a significant difference between the tumor and BPH ROIs, although the two-component model yields the lowest p-value.

Fig 3 shows examples of calculated $SF_{slow}$ and ADC maps for PZ and non-PZ tumors and BPH, as well as corresponding histology slides for the PZ and non-PZ tumors and a T2-weighted image for the BPH case. $SF_{slow}$ (for both the two-component and bi-exponential models) and ADC yield good tumor conspicuity for both tumor cases. In the BPH case, the lesion is visible both on the $SF_{slow}$ maps and the ADC map, although with a slightly lower contrast than the tumors.

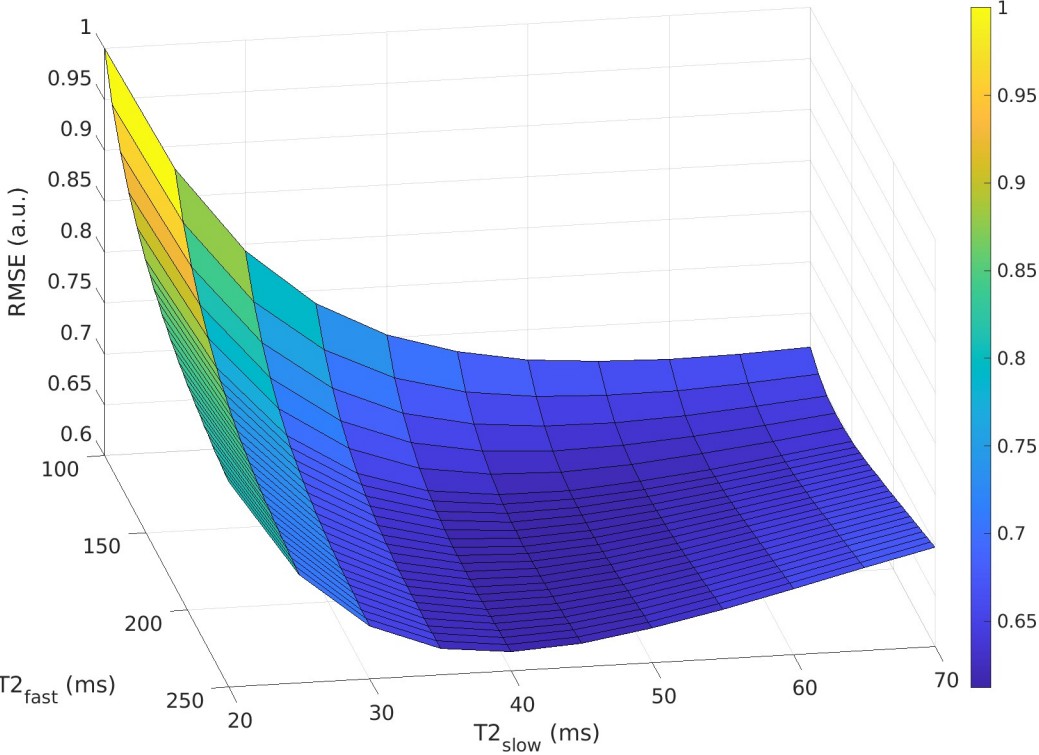

**Fig 1. Surface curve showing the total root-mean-square error (RMSE) from fitting the two-component model with a range of T2$_{slow}$ and T2$_{fast}$ values for the training set, scaled so that the highest RMSE equals 1.** The total RMSE is at a minimum for T2$_{slow}$ = 45 ms and T2$_{fast}$ = 180 ms.

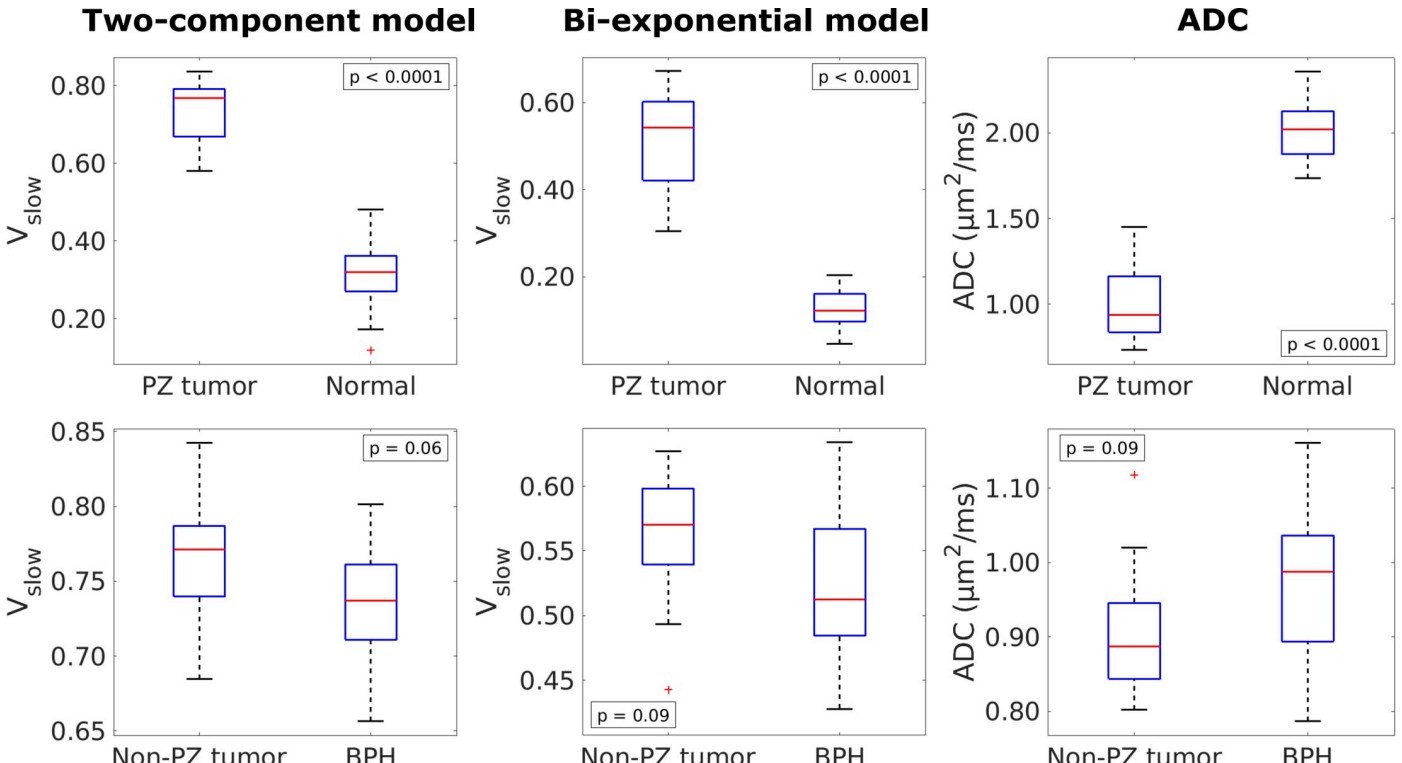

**Fig 2. Box plots showing the distribution of mean SF$_{slow}$ (from the two-component and bi-exponential models) and ADC of the different ROIs.** Upper row: PZ tumors (n = 24) compared to normal tissue (n = 24). Bottom row: Non-PZ tumors (n = 13) compared to BPH (n = 14). $p<0.006$ was considered statistically significant.

ROC curves and results from the ROC analysis are shown in Fig 4 and Table 2, respectively, and show that SF$_{slow}$ (for both the two-component and bi-exponential model) and ADC have very good diagnostic performance. However, although the sensitivity, specificity and area under the ROC curve (AUC) are very similar for all metrics, SF$_{slow}$ from the two-exponential model yields slightly higher values than the other two, which are nearly identical to each other. Note the very different optimal threshold values for SF$_{slow}$ from the two-exponential and bi-exponential models of 0.67 and 0.42, respectively. This means that $\geq$67% and 42% of the voxels should be in the SF$_{slow}$ components to be classified as tumor tissue. In the case of ADC, the value needs to be under the respective optimal threshold to be classified as tumor.

In Fig 5, the mean SF$_{slow}$ and ADC from the tumor ROIs are plotted as a function of Gleason Grade Group. The Spearman correlation coefficient was found to be $\rho = 0.370$ ($p = 0.040$) and $\rho = 0.499$ ($p = 0.004$) for SF$_{slow}$ for the two-component and bi-exponential model, respectively, and $\rho = -0.490$ ($p = 0.005$) for ADC.

## Discussion

In this study, 62 patients with prostate cancer and 14 patients with BPH underwent combined T2-DWI. From this, signal fractions were estimated using a two-component model based on both T2 and ADC dependence. The purpose was to investigate the diagnostic potential of this model in comparison with results from conventional diffusion models. Our results show that SF$_{slow}$ from the two-component model is higher in PZ tumors than in normal tissue, and in non-PZ tumors than in BPH, although only significant in the former case. SF$_{slow}$ shows good diagnostic properties and a fair correlation with tumor aggressiveness.

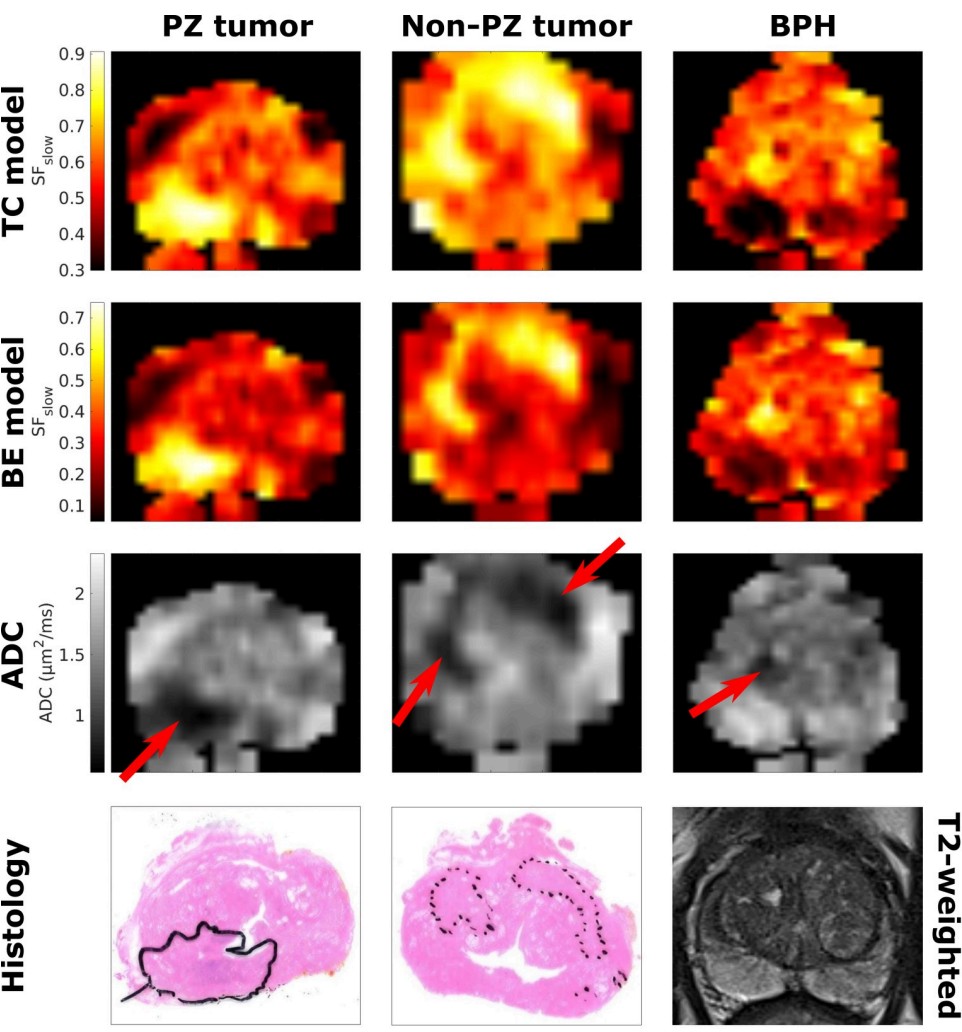

**Fig 3. Examples of estimated value maps for three different patients.** $SF_{slow}$ (from the two-component (TC) and bi-exponential (BE) models) maps and corresponding ADC map are shown for a PZ tumor patient, a non-PZ tumor patient and a BPH patient, respectively. For the tumor patients, histology slides are also shown for comparison, while a corresponding T2-weighted image is shown in the BPH case as histology was not available for this patient. The lesions are denoted on the ADC maps with red arrows. On the histology slides, the PZ tumor is denoted with a solid line, while the non-PZ tumors are denoted with dotted lines. Example maps from more cases can be found in the S2 Fig.

Global T2 optimization was performed both to reduce the number of free variables and to isolate components with distinct, paired ADC and T2 values. Comparing the results with a model fit with the T2 values as free variables (see S1 Table) showed the benefit of optimizing the T2 values in advance. To further reduce the number of free variables, the ADCs were adapted from a bi-exponential model [19]. The signal fraction estimates for the slow component in that paper were markedly lower than our $SF_{slow}$ estimates from the two-component model, but closer to the values obtained with our bi-exponential model, indicating a dependence of $SF_{slow}$ on T2. This implies that the two-component and bi-exponential models do not isolate identical subvoxel populations of water molecules, because the added T2 dependence actively tunes the diffusion signal from the prostate tissue, in agreement with other studies performed with combined T2-DWI [12–14]. A previous three-component model based on combined T2-DWI estimated the T2 values of the epithelium, stroma and lumen to be 50 ms, 80

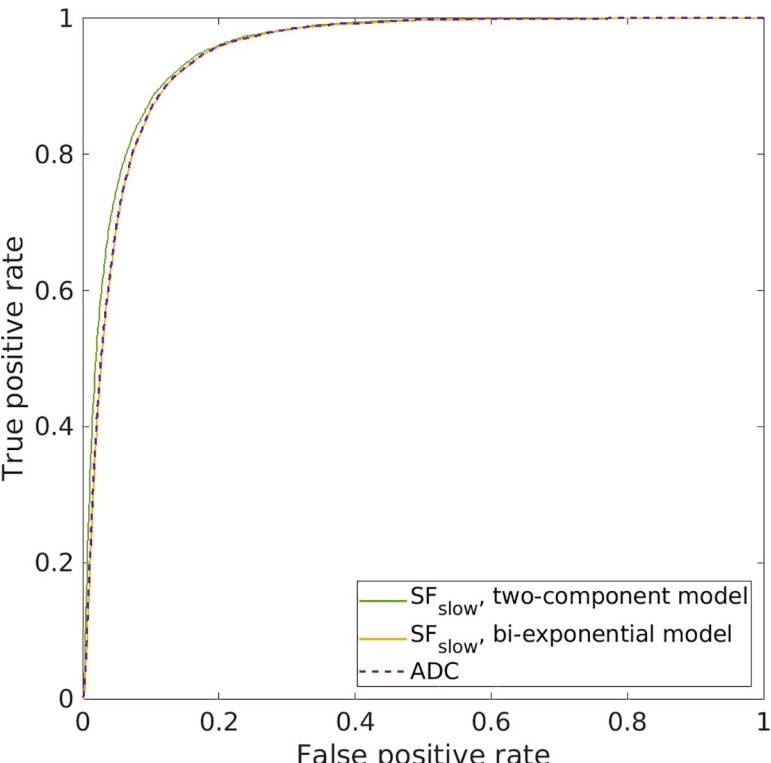

**Fig 4. ROC curves for SF$_{slow}$ (for the two-component model in green and for the bi-exponential model in yellow) and ADC (in dashed purple).** Voxels within tumor ROIs (n = 6569) were compared with voxels outside (n = 254,112).

ms and 665 ms, respectively [15], which would suggest that our slow component is approximately equivalent to the epithelium, whereas the fast component is a mixture of stroma and lumen.

The two-component model performed similarly to the bi-exponential model and the mono-exponential ADC. All three models showed a highly significant difference between PZ tumors and normal tissue, and while no significant differences were found between non-PZ tumors and BPH, the two-component model performed slightly better than the others in that case. In the ROC analysis, where all three models showed an excellent diagnostic performance, the two-component model also performed marginally better than the other two. However, the bi-exponential model and ADC showed a somewhat better correlation with tumor aggressiveness, although none of them significant. Interestingly, in all the analyses, the bi-exponential model and ADC both show very similar results that are slightly different from the two-component model, which could suggest that our model extracts additional information from the underlying tissue compared to the other models. Although our two-component model

**Table 2. Summary of Receiver Operating Characteristics (ROC) results.**

| Model | AUC | Optimal threshold | Sensitivity (%) | Specificity (%) |
|---|---|---|---|---|
| SF$_{slow}$, two-component model | 0.956 | 0.67 | 92.6 | 85.8 |
| SF$_{slow}$, bi-exponential model | 0.949 | 0.42 | 92.3 | 85.4 |
| ADC (μm$^2$/ms) | 0.949 | 1.17 | 92.3 | 85.4 |

AUC = area under curve, SF = signal fraction, ADC = apparent diffusion coefficient.

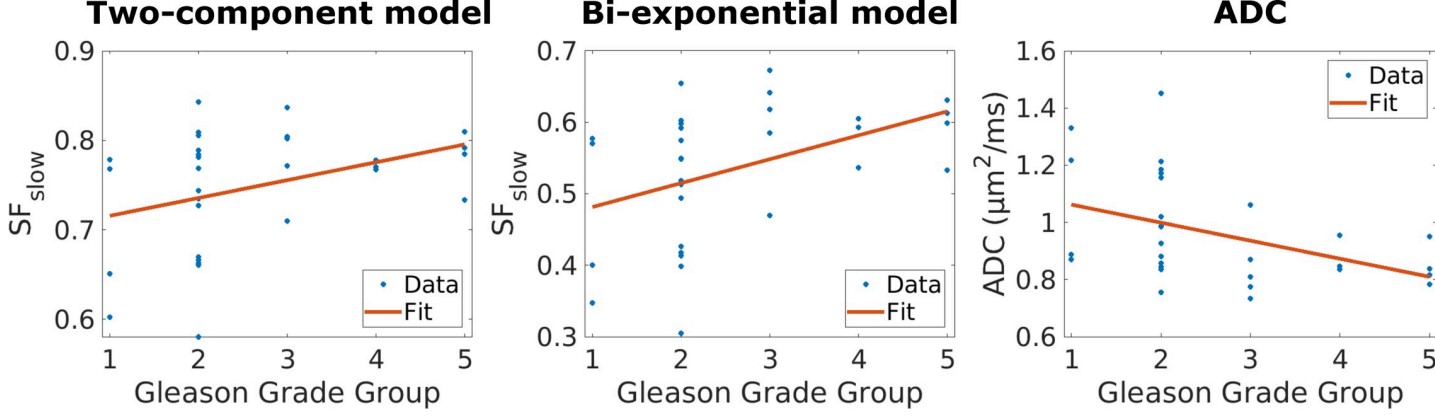

**Fig 5. Mean tumor ROI values plotted as a function of Gleason Grade Group (blue dots) for SF$_{slow}$ (from the two-component and bi-exponential models) and ADC.** The red line represents the least-square fit to the data.

performed similarly to conventional models, and not significantly better, the method shows diagnostic promise and should be further optimized and investigated in a larger number of patients, in order to more rigorously evaluate its ability to predict tumor aggressiveness.

One reason for the similar performance between the two-component and bi-exponential models and ADC could be that the tumor, normal tissue and BPH ROIs used in this study were all delineated on ADC maps. Delineating the ROIs based on MR images and not histopathology of prostatectomy specimens can give a bias in the results, such that it is more difficult to see improvements in other models if ADC is used as a reference. Nevertheless, we chose to use radiology for the delineation because including only prostatectomy patients would give a bias towards highly aggressive cancer, as well as reducing the number of patients available. Another reason that the models perform so similarly could be that the bi-exponential model and ADC are calculated at the longest TE. Therefore, they are also implicitly influenced by the T2 relaxation of the tissue to a higher degree than if the shorter TE was used. However, we chose to use TE = 73 ms since this was TE closest to the one used in [19]. A calculation with TE = 55 ms is carried out in the S1 Table for comparison.

We show that SF$_{slow}$ from the two-component model has diagnostic potential in prostate cancer. Some might argue that a more complex model would give a better representation of the underlying tissue microstructure, but our focus was to apply a model that would be feasible in a clinical setting, where time is a limiting factor. Our T2-DWI protocol had a comparable acquisition time to that of a standard clinical prostate DWI sequence, and there are only two variables to be estimated, given fixed ADCs and T2 values of each signal component. However, these values should be further optimized by exploring a wider range of TEs and b-values, in order to potentially increase the diagnostic performance of the method. Furthermore, since the main focus of our work was to investigate clinical feasibility of the two-component model, we did not perform a thorough evaluation of repeatability and reproducibility, which should also be addressed in the future.

Our study had some limitations. Firstly, the images were not corrected for geometric distortions caused by the EPI sequence, although no severe distortions were noted during visual inspection of the images. Secondly, the sequence parameters of the combined T2-DWI were introduced to see whether there were any effects of varying the TE in DWI, and it was made as short as possible to fit in at the end of a clinical protocol, with TEs and b-values close to clinical DWI parameters. The low number and short range of these parameters limit the sensitivity of the T2 values and ADC of the components. At last, because a standard vendor DWI sequence

was used for the combined T2-DWI, the acquisition at the different TEs had different diffusion times ($\Delta$ and $\delta$), which can also affect the apparent TE dependence of ADC [21]. This should be addressed when designing new combined T2-DWI protocols.

In conclusion, signal fraction estimates from a two-component model based on combined T2-DWI can differentiate between PZ tumors and normal prostate tissue and show potential for prostate cancer diagnosis. The model performed similarly to conventional diffusion models. However, the method should be further optimized for clinical purposes and investigated in a larger number of patients.

## Supporting information

**S1 File. Minimal underlying data.** The values (for individual patients/ROIs/voxels) behind the tables and figures, and the statistical measures reported.
(XLSX)

**S1 Fig. Patient and ROI selection.**
(PDF)

**S2 Fig. Examples of estimated value maps for three different patients.** $SF_{slow}$ (from the two-component (TC) and bi-exponential (BE) models) maps and corresponding ADC map are shown for a PZ tumor patient, a non-PZ tumor patient and a BPH patient, respectively. For the tumor patients, histology slides are also shown for comparison, while a corresponding T2-weighted image is shown in the BPH case as histology was not available for this patient. The lesions are denoted on the ADC maps with red arrows. On the histology slides, the tumors are denoted with dotted lines.
(DOCX)

**S1 Appendix. Supplementary methods.**
(DOCX)

**S1 Table. Median [95% CI] of the mean region of interest (ROI) values calculated from the different models and parameters.** CI = confidence interval, PZ = peripheral zone, BPH = benign prostatic hyperplasia, SF = signal fraction, ADC = apparent diffusion coefficient, TE = echo time. Note—For PZ tumors, tumor ROIs (n = 24) were compared with normal ROIs (n = 24), while for non-PZ tumors, tumor ROIs (n = 13) were compared with BPH ROIs (n = 14). Units are given in parentheses, except for $SF_{slow}$ which is unitless. A p-value<0.0019 was considered statistically significant, and the significant results are highlighted in bold.
(DOCX)

## Acknowledgments

We thank Sverre Langørgen for help with radiologic assessment. We also want to thank Kjerstin Olaussen, Torill E. Sjøbakk, Mohammed R. S. Sunoqrot and Daniel Chen Billdal for help with data collection and organization.

## Author Contributions

**Conceptualization:** Ingrid Framås Syversen, Mattijs Elschot, Tone Frost Bathen, Pål Erik Goa.

**Formal analysis:** Ingrid Framås Syversen.

**Funding acquisition:** Mattijs Elschot, Tone Frost Bathen, Pål Erik Goa.

**Investigation:** Ingrid Framås Syversen, Elise Sandsmark.

**Methodology:** Ingrid Framås Syversen, Pål Erik Goa.

**Project administration:** Tone Frost Bathen, Pål Erik Goa.

**Resources:** Pål Erik Goa.

**Software:** Ingrid Framås Syversen.

**Supervision:** Pål Erik Goa.

**Validation:** Elise Sandsmark, Helena Bertilsson.

**Visualization:** Ingrid Framås Syversen.

**Writing – original draft:** Ingrid Framås Syversen.

**Writing – review & editing:** Ingrid Framås Syversen, Mattijs Elschot, Elise Sandsmark, Helena Bertilsson, Tone Frost Bathen, Pål Erik Goa.

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
