## [Decision Letter · Decision Letter 0]

8 Apr 2021

PONE-D-21-06805

Exploring the diagnostic potential of adding T2 dependence in diffusion-weighted MR imaging of the prostate

PLOS ONE

Dear Dr. Syversen,

Thank you for submitting your manuscript to PLOS ONE. After careful consideration, we feel that it has merit but does not fully meet PLOS ONE’s publication criteria as it currently stands. Therefore, we invite you to submit a revised version of the manuscript that addresses the points raised during the review process.

We look forward to receiving your revised manuscript.

Kind regards,

Pascal A. T. Baltzer, M.D.

Academic Editor

PLOS ONE

Journal Requirements:

Additional Editor Comments:

Dear authors!

It has been difficult to find reviewers so I proceed with one review collected and my own assessment. In line with R1, I see the need for multiple clarifications regarding patient selection, the exact details of the reference standard and correlation of imaging and histology. Further, your reporting could be read as simple ADC having exactly the same diagnostic ability than the suggested more complex models. The only clinically valid interpretation could then be that basic ADC measurements are sufficient to diagnose prostate cancer. Whether your approach provides e.g. an edge to predict Gleason Grade /ISUP of the PCa lesions (an interesting field of research) is another question that might at least be explored.

Reviewers' comments:

Reviewer's Responses to Questions

**Comments to the Author**

1. Is the manuscript technically sound, and do the data support the conclusions?

Reviewer #1: Partly

2. Has the statistical analysis been performed appropriately and rigorously? 

Reviewer #1: Yes

3. Have the authors made all data underlying the findings in their manuscript fully available?

Reviewer #1: No

4. Is the manuscript presented in an intelligible fashion and written in standard English?

Reviewer #1: Yes

5. Review Comments to the Author

Reviewer #1: Abstract: Ok.

Intro: Ok. Might want to mention that also DCE sequences are used, as they are still part of the PIRADS and the use of bp instead of mp prostate MRI is still under discussion.

M&M/Results

I am a bit puzzled by lesion selection. Were areas of prostatitis included? Or only normal prostate/BPH/cancer? Were the ROIs correlated with histology to ensure the type of tissue?

Overall, while the authors are very careful in describing the technical parts, only few details are given on the standard of reference. Most important, it is not clear whether a standard of reference was available for normal/BPH areas, which should be mandatory in this kins of studies.

The number of patients is given, but not the number of lesions. Was only one pathologic and one normal area per patient considered?

It seems that ADC performs as good if not better than the suggested models. I wonder whether these results might somehow be related to the fact that ADC was, from what I can understand, to define the ROI.

Overall, I am not only concerned by the small number of cases included, but also by the case and lesion selection - as the authors also actually discuss in the text. While the authors discuss this limitation as minor, I am worried it might have actually influenced the results of the analysis.

I would strongly suggest to improve the case included and improve the ROI choice/methods to confirm the results.

6. PLOS authors have the option to publish the peer review history of their article (what does this mean?). If published, this will include your full peer review and any attached files.

Reviewer #1: No

---

## [Author Response · Author response to Decision Letter 0]

30 Apr 2021

Response to reviewers

We thank the reviewers for providing useful comments that led us to improve our manuscript, especially regarding patient and ROI selection. We have tried our best to clarify what was unclear, and in light of the feedback we have also improved some steps of the methods and re-run the analyses. Below are our detailed responses to the comments from each reviewer.

Reviewer #1 comments:

Intro:

1. Might want to mention that also DCE sequences are used, as they are still part of the PIRADS and the use of bp instead of mp prostate MRI is still under discussion.

• We have now rewritten and included this in our Introduction: “The Prostate Imaging-Reporting and Data System (PI-RADS) guidelines are used to detect clinically significant cancer, based on a combination of DWI, T2-weighted imaging and also dynamic contrast-enhanced (DCE) MRI” (page 3, lines 51-54).

M&M/Results:

2. I am a bit puzzled by lesion selection. Were areas of prostatitis included? Or only normal prostate/BPH/cancer?

• Only tumor, normal tissue and BPH were included in the ROIs, not prostatitis. For more information about the patient and ROI selection process, we have made a new Supporting information figure, “S1 Fig. Patient and ROI selection”, with a flow chart that summarizes the process.

3. Were the ROIs correlated with histology to ensure the type of tissue?

• We thank you for bringing up your concern about the correlation between imaging (ROIs) and biopsy/histology. In light of your comments, we have chosen to revise this step of the ROI delineation process. Earlier, we only had information about the highest Gleason Grade Group reported for the whole prostate, and this was then assigned to the ROI delineated based on the clinical radiologist’s PI-RADS reporting. However, we have now gone through the biopsy and histology reports for all the included patients, and have closely matched the ROI with Gleason Grade Group reported from that location from the biopsy/histology, and cross-checked the ROI location with the whole-mount histology slides if available for the patient. We have therefore added a sentence to clarify this in the “Materials and methods” > “ROI delineation” section: “All tumor ROI locations were confirmed to be cancer by matching with biopsy reports in the patient journal, and cross-checked with histology slides if available (n=29).” (page 7, lines 137-139). This resulted in the following changes to the ROIs and analysis results:

i. No new tumor ROIs were drawn, but for some patients a part of the ROI was removed in order to match the biopsy/histology reporting.

ii. In some cases, the Gleason Grade Group was changed in order to match the location of the tumor ROI (e.g. we no longer used the highest Gleason Grade Group from the whole prostate).

iii. Some patients (n=4) initially received other treatment (e.g. active surveillance), but when going through the journals again, we discovered that they eventually underwent prostatectomy (RARP). Gleason Grade Group and treatment were updated in Table 1 (page 5, lines 95-104) to reflect this.

iv. One patient was excluded from the analyses, because the tumor ROI location described by the PI-RADS did not show a positive biopsy at that location (the biopsy was positive at another location not PI-RADS described).

v. In result, implementing the above-mentioned changes in our analyses produced slightly different numeric values in tables and figures (and lower p-values), but the differences were not large enough to change the level of statistical significance of the results nor the conclusions of our work.

4. Overall, while the authors are very careful in describing the technical parts, only few details are given on the standard of reference. Most important, it is not clear whether a standard of reference was available for normal/BPH areas, which should be mandatory in this kins of studies.

• We have expanded the section “Materials and methods” > “ROI delineation” to also include information about the reference standards for the ROI delineation: “The delineation was performed based on the clinical radiologist’s PI-RADS annotation and the tumor ROI was characterized by focal low ADC under (or around) 1000 µm2/ms.” (page 7, lines 135-37), “For the PZ tumor patients, one normal tissue ROI was also delineated, characterized by visually high ADC in the PZ (around 2000 µm2/ms).” (page 7, lines 139-140), “For each BPH patient, one ROI was manually delineated of one or more proliferative BPH nodule(s) with visually low ADC (under/around 1000 µm2/ms) in the non-PZ, also visible as a nodule on T2-weighted images.” (page 7, lines 142-144).

5. The number of patients is given, but not the number of lesions. Was only one pathologic and one normal area per patient considered?

• For the tumor and normal tissue ROIs, only one area was considered in each ROI. However, the BPH ROIs can contain up to several nodules. We have tried to clarify this by adding to the “Materials and methods” > “ROI delineation” section: “For both the tumor and normal tissue ROIs, respectively, only one tumor lesion or normal area were considered per ROI. For each BPH patient, one ROI was manually delineated of one or more proliferative BPH nodule(s)” (page 7, lines 140-143).

6. It seems that ADC performs as good if not better than the suggested models. I wonder whether these results might somehow be related to the fact that ADC was, from what I can understand, to define the ROI. Overall, I am not only concerned by the small number of cases included, but also by the case and lesion selection – as the authors also actually discuss in the text. While the authors discuss this limitation as minor, I am worried it might have actually influenced the results of the analysis. I would strongly suggest to improve the case included and improve the ROI choice/methods to confirm the results.

• We understand the reviewer’s concerns, and based on the comments we have improved patient/ROI selection and clarified our methods as described. However, it is not possible to address all of the concerns in our current patient cohort. The reviewer is correct that the ROIs were delineated on ADC maps, and correctly points out that we already discuss this: “One reason for the similar performance between the two-component and bi-exponential models and ADC could be that the tumor, normal tissue and BPH ROIs used in this study were all delineated on ADC maps. Delineating the ROIs based on MR images and not histopathology of prostatectomy specimens can give a bias in the results, such that it is more difficult to see improvements in other models if ADC is used as a reference. Nevertheless, we chose to use radiology for the delineation because including only prostatectomy patients would give a bias towards highly aggressive cancer, as well as reducing the number of patients available.” (page 14, lines 304-311). The reviewer is also concerned about the small number of cases included. From our point of view, these two factors (ROIs delineated from ADC vs. small number of cases) are a trade-off in our study. As we point out in the text above, we chose to delineate the ROIs based on radiology annotations, and not only from whole-mount histology after RARP because that would reduce the total number of patients in the study (in our case, that would only be the 29 patients who underwent prostatectomy, see Table 1 (page 5, lines 94-103)). Unfortunately, we are not able to include any more cases than what we already have, as we have already included all available prostate cancer patients that meet our inclusion criteria, and the data collection of this cohort is completed. We agree that the method should be further investigated in a larger number of patients, as we point out in the Discussion section: “(…) the method shows diagnostic promise and should be further optimized and investigated in a larger number of patients, in order to more rigorously evaluate its ability to predict tumor aggressiveness.” (page 14, lines 301-303), and in our conclusion: “(…) the method should be further optimized for clinical purposes and investigated in a larger number of patients.” (page 15, lines 342-343). We think that the methods are relevant to researchers working in this field, and results presented in this manuscript can be used as a starting point to further optimize the imaging protocol and parameters of the two-component model in a larger cohort.

Editor comments:

1. In line with R1, I see the need to multiple clarifications (a) regarding patient selection, (b) the exact details of the reference standard and (c) correlation of imaging and histology.

• (a) We have tried to clarify the patient selection process by making a new Supporting information figure, “S1 Fig. Patient and ROI selection”, with a flow chart that summarizes the process. We now refer to this figure in the “Materials and methods” > “Patients” section: “An overview of the patient and case selection process can be found in the Supporting information (S1 Fig).” (page 4, lines 89-90), and we have also added a line about inclusion criteria for clarification: “The inclusion criteria were tumor in any prostate zone that both had a PI-RADS and a location-matched Gleason score.” (page 4, lines 86-88).

• (b) As reviewer #1 also raised the question about reference standards, we refer to our answer to R1’s comment 4.

• (c) This question was also raised by reviewer #1, and we have improved this step accordingly. For a detailed description of correlation between imaging and histology, we refer to our answer to R1’s comment 3.

2. Further, your reporting could be read as simple ADC having the same diagnostic ability than the suggested more complex models. The only clinically valid interpretation could then be that basic ADC measurements are sufficient to diagnose prostate cancer. Whether your approach provides e.g. an edge to predict Gleason Grade/ISUP of the PCa lesions (an interesting field of research) is another question that might at least be explored.

• Although the two-component model, bi-exponential model and ADC performed similarly, the diagnostic performance of ADC was not the research question of our manuscript, and we do therefore not quite agree that this should be the conclusion. ADC is already well-established in the PI-RADS guidelines as part of the diagnostic criteria. However, there also exist other, more advanced diffusion models, and we cannot rule out that these might perform better than ADC. Our research question was to investigate the diagnostic performance of the two-component model, and we found it to be just as good as the conventional models – which is also then our conclusion. We do not make any claims about the diagnostic performance of the two-component model that is not supported by the data. However, as you point out, it will be an interesting question whether this model can provide extra information about tumor aggressiveness, and we also mention in our Discussion section that this should be further explored: “(…) in all the analyses, the bi-exponential model and ADC both show very similar results that are slightly different from the two-component model, which could suggest that our model extracts additional information from the underlying tissue compared to the other models. Although our two-component model performed similarly to conventional models, and not significantly better, the method shows diagnostic promise and should be further optimized and investigated in a larger number of patients, in order to more rigorously evaluate its ability to predict tumor aggressiveness.” (page 13-14, lines 296-303), and “(…) fixed ADCs and T2 values of each signal component. However, these values should be further optimized by exploring a wider range of TEs and b-values, in order to potentially increase the diagnostic performance of the method. Furthermore, since the main focus of our work was to investigate clinical feasibility of the two-component model, we did not perform a thorough evaluation of repeatability and reproducibility, which should also be addressed in the future.” (page 14-15, lines 322-327). The methods and results presented in this manuscript can then be used as a starting point to optimize the imaging protocol and parameters of the two-component model in a larger cohort.

---

## [Editor Report · Decision Letter 1]

17 May 2021

Exploring the diagnostic potential of adding T2 dependence in diffusion-weighted MR imaging of the prostate

PONE-D-21-06805R1

Dear Dr. Syversen,

We’re pleased to inform you that your manuscript has been judged scientifically suitable for publication and will be formally accepted for publication once it meets all outstanding technical requirements.

Kind regards,

Pascal A. T. Baltzer, M.D.

Academic Editor

PLOS ONE

---

## [Editor Report · Acceptance letter]

19 May 2021

PONE-D-21-06805R1 

Exploring the diagnostic potential of adding T2 dependence in diffusion-weighted MR imaging of the prostate 

Dear Dr. Syversen:

I'm pleased to inform you that your manuscript has been deemed suitable for publication in PLOS ONE. Congratulations! Your manuscript is now with our production department. 

Kind regards, 

on behalf of

Dr. Pascal A. T. Baltzer 

Academic Editor

PLOS ONE